# Bridging Compressed Image Latents and Multimodal Large Language Models

**Chia-Hao Kao[1]   Cheng Chien[2]   Yu-Jen Tseng[2]   Yi-Hsin Chen[2]   Alessandro Gnutti[1]
Shao-Yuan Lo[3]   Wen-Hsiao Peng[2]   Riccardo Leonardi[1]**
[1]University of Brescia, Italy    [2]National Yang Ming Chiao Tung University, Taiwan
[3]Honda Research Institute USA

## Abstract

This paper presents the first-ever study of adapting compressed image latents to suit the needs of downstream vision tasks that adopt Multimodal Large Language Models (MLLMs). MLLMs have extended the success of large language models to modalities (e.g. images) beyond text, but their billion scale hinders deployment on resource-constrained end devices. While cloud-hosted MLLMs could be available, transmitting raw, uncompressed images captured by end devices to the cloud requires an efficient image compression system. To address this, we focus on emerging neural image compression and propose a novel framework with a lightweight transform-neck and a surrogate loss to adapt compressed image latents for MLLM-based vision tasks. Given the huge scale of MLLMs, our framework excludes the entire downstream MLLM except part of its visual encoder from training our system. This stands out from most existing coding for machine approaches that involve downstream networks in training and thus could be impractical when the networks are MLLMs. The proposed framework is general in that it is applicable to various MLLMs, neural image codecs, and multiple application scenarios, where the neural image codec can be (1) pre-trained for human perception without updating, (2) fully updated for joint human and machine perception, or (3) fully updated for only machine perception. Extensive experiments on different neural image codecs and various MLLMs show that our method achieves great rate-accuracy performance with much less complexity.

## 1 Introduction

Large Language Models (LLMs) (Touvron et al., 2023b;a) have demonstrated impressive abilities in various Natural Language Processing (NLP) tasks. Building upon their success, the recent surge of research on Multimodal Large Language Models (MLLMs) extends LLM's abilities to modalities beyond languages, particularly images, opening up promising opportunities in various applications (Achiam et al., 2023; Cha et al., 2024; Lin et al., 2024; Zhang et al., 2024a). MLLMs have shown surprising capability for many vision tasks such as classification (Zhu et al., 2024b), image captioning (Zhang et al., 2024a), Visual Question Answering (VQA) (Cha et al., 2024; Lin et al., 2024), and meme interpretation (Achiam et al., 2023). These models excel in unseen tasks through instruction following or in-context learning, which is impossible for traditional vision networks.

However, the massive scale of MLLMs, often comprising billions of parameters, poses significant challenges for deployment on resource-constrained end devices. While computation can be offloaded to the cloud, transmitting images to cloud-hosted MLLMs becomes necessary. In this case, efficient image compression techniques are crucial to reducing the required transmission bit-rate. Without compression, transmitting raw images incurs significant costs, particularly at scale with numerous users. Our study shows that simply feeding the decoded image, generated by a fixed image codec trained for human perception, into an MLLM (Figure 1 (a)) substantially degrades task performance, particularly when the image is coded at low rates. This highlights the critical need for efficient image compression that considers the requirements of downstream MLLM-based vision tasks.

To the best of our knowledge, there have been no attempts to tackle image compression specifically for MLLMs. While many prior works address image compression for machine vision, commonly referred to as coding for machines (Le et al., 2021b; Chamain et al., 2021; Matsubara et al., 2022;

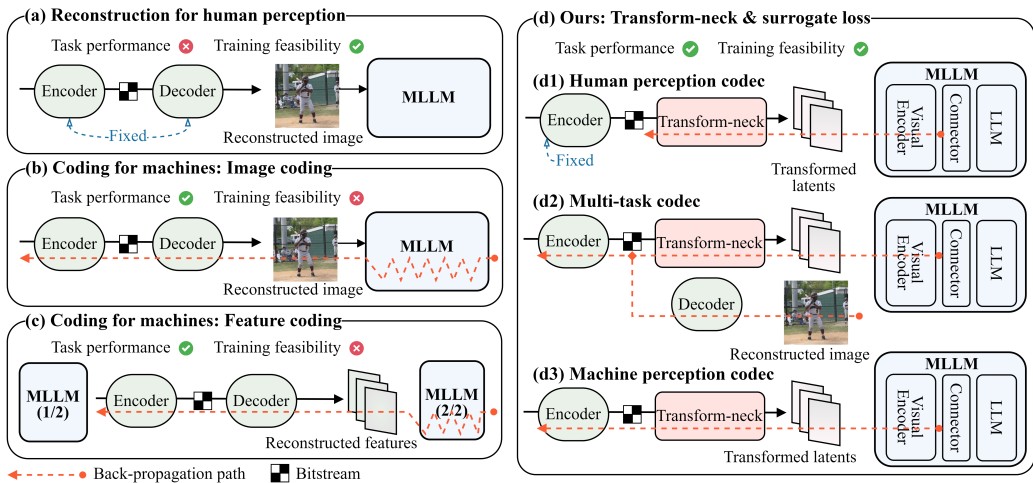

Figure 1: On the left is inadequate frameworks for image compression for MLLMs, where the image codec is trained for (a) human perception, (b) the downstream task network, or (c) compressing the intermediate features of the task network. On the right is the proposed transform-neck and surrogate loss under three distinct scenarios, with the image codec (d1) pre-trained for human perception, (d2) updated for joint human and machine perception, or (d3) updated for machine perception.

Liu et al., 2022b), these approaches cannot be directly applied to MLLMs. Two common approaches to coding for machines are image coding and feature coding. The image coding approaches (Le et al., 2021b;a) optimize the image codec for specific downstream tasks and/or networks (Figure 1 (b)), while the feature coding approaches (Ding et al., 2024) divide the task network into two parts and focus on compressing the intermediate features (Figure 1 (c)). However, both approaches face the same issue: the training process becomes challenging when one needs to back-propagate a training objective through a massive MLLM to train the neural image codec. In practice, the billion-scale parameters of MLLMs make the existing coding for machine methods inapplicable.

In this paper, we propose the first neural image compression system for MLLM-based vision tasks that enables compressed latents to suit the needs of downstream MLLMs. Notably, it is not our objective to develop a new image codec specifically for MLLM-based tasks. Instead, our method involves a lightweight transform-neck and a novel surrogate loss. The transform-neck adapts the compressed image latents of an off-the-shelf neural image codec to match the intermediate features of the visual encoder–i.e. a component of MLLMs–bypassing the needs for full image reconstruction and reducing computational complexity. Our proposed surrogate loss function, which combines the cross-entropy and distillation terms, enables our system to be trained by back-propagating solely through the visual encoder, thus eliminating the need for back-propagation through the entire MLLM.

The proposed method is general in that it is applicable to different neural image codecs under various application scenarios. First, if the downstream applications prioritize the image reconstruction quality for human interaction, our method can work with an off-the-shelf image codec trained for human perception (Figure 1 (d1)). Without any modification or re-training of the codec, our method adapts the compressed image latents while maintaining the same image reconstruction quality. Second, when allowing the image codec to be updated, we propose a multi-task training strategy that optimizes the codec for both human and machine perception (Figure 1 (d2)). This significantly improves MLLM performance at the cost of a marginal drop in the image's reconstruction quality. Finally, we consider an extreme setting in which the applications prioritize machine perception over image reconstruction. In this case, the encoder and the transform-neck are jointly optimized for the MLLM systems exclusively (Figure 1 (d3)). The main contributions of this work are summarized as follows:

- It marks the first exploration into the field of neural image coding for MLLMs.
- The proposed transform-neck adapts the compressed image latents to downstream MLLMs, avoiding the need for image reconstruction and thus saving computational complexity.
- The proposed surrogate loss leverages the visual encoder to update the system, avoiding back-propagating the task loss through the heavy MLLM.
- The proposed framework is broadly applicable to a wide range of neural image codecs and MLLMs, regardless of their architectures.

- It is able to accommodate various application scenarios that involve human perception, machine perception, or both.

Last but not least, the transform-neck trained with our surrogate loss exhibits a degree of universality, since it is readily applicable to multiple MLLMs that share the same visual encoder, without the need for retraining. Our method achieves (1) up to 60-80% bit-rate reductions under the same recognition accuracy over existing image codecs (e.g. ELIC (He et al., 2022) and VVC intra coding (Bross et al., 2021)) (Sections 4.2 and A.2) and (2) a nearly 95% reduction in decoding kMAC/pixel as compared to performing full image reconstruction followed by enhancing the reconstructed image for MLLM-based tasks (Section 4.4). Our system can be successfully trained under various application scenarios on one RTX 4090 with 24GB of memory. This is not possible when the entire MLLM is involved in the training process.

## 2 RELATED WORKS

### 2.1 MULTIMODAL LARGE LANGUAGE MODELS

In recent years, there has been a surge of interest in MLLMs following the impressive demonstration of LLM's ability in the NLP field (Touvron et al., 2023b;a; Jiang et al., 2023). Many have sought to extend the success of these models from text to other modalities, particularly images, and several works have shown their effectiveness on various tasks, such as image captioning (Li et al., 2023b; Lin et al., 2024; Liu et al., 2023a), VQA (Cha et al., 2024; Zhang et al., 2024a), Referring Expression Comprehension (REC) (Chen et al., 2023a; Peng et al., 2024; Zhang et al., 2024b), few-shot classification (Yu et al., 2024; Zhu et al., 2024b), action anticipation (Mittal et al., 2024).

Most existing MLLM approaches use a visual encoder to process the input image data, and then introduce a connector to bridge the image features to the tokens understandable by the LLM. Earlier works adopt simpler connector designs, such as linear projectors (Chen et al., 2023a; Liu et al., 2023a), while subsequent works (Li et al., 2023b; Cha et al., 2024; Zhang et al., 2024a) have refined upon the design for both performance and complexity. Furthermore, the entire MLLM can be further fine-tuned to enhance its capabilities through instruction tuning (Liu et al., 2023a; Zhu et al., 2024a).

A notable aspect of the MLLMs is their reliance on existing pre-trained visual encoders in their systems, with CLIP (Radford et al., 2021) visual encoder being a very common choice for a large number of methods (Cha et al., 2024; Chen et al., 2023a; Zhang et al., 2024a; Zhu et al., 2024b; Lin et al., 2024; Li et al., 2023b). Trained on large image-text pair data, the CLIP visual encoder offers the feature space that combines language and image modalities in a sense, making it a desirable feature for MLLMs. Notably, all the existing works on MLLMs do not consider the scenarios where image compression is present, which is a significant departure from our work. We note that some approaches (Shi et al., 2024; Li et al., 2024) perform token reduction to minimize the inference cost of the downstream MLLMs. These techniques are orthogonal to and can be combined with our method (see Section A.6 for more discussions).

### 2.2 IMAGE CODING FOR MACHINES

Neural image compression systems have made significant progress in the past few years. As a matter of fact, several works (He et al., 2022; Liu et al., 2023b) have even outperformed the traditional codecs such as intra coding in VVC (Bross et al., 2021). However, these methods primarily focus on the quality of reconstructed images for human perception. Coding for machines, in contrast, targets downstream machine vision over human perception, and it has attracted increasing attention recently.

A common approach simply involves training the compression system for a predefined target downstream computer vision task (Le et al., 2021b;a; Wang et al., 2022), enabling the reconstructed image to be suitable for machine vision, albeit potentially sacrificing perceptual quality. Conversely, Chamain et al. (2021) tune the task network to better process the compressed images, while Chen et al. (2023b) leverage prompt-tuning method on Transformer-based codecs to boost performance on multiple tasks. Also, with the trend of the new JPEG AI learning-based image coding standard (Ascenso et al., 2023), some methods (Liu et al., 2022a; 2021; Mei et al., 2021; Singh et al., 2020) utilize the compressed image latents instead of the reconstructed image for recognition through bridging the latents to task network. On the other hand, (Ding et al., 2024) directly compress the intermediate

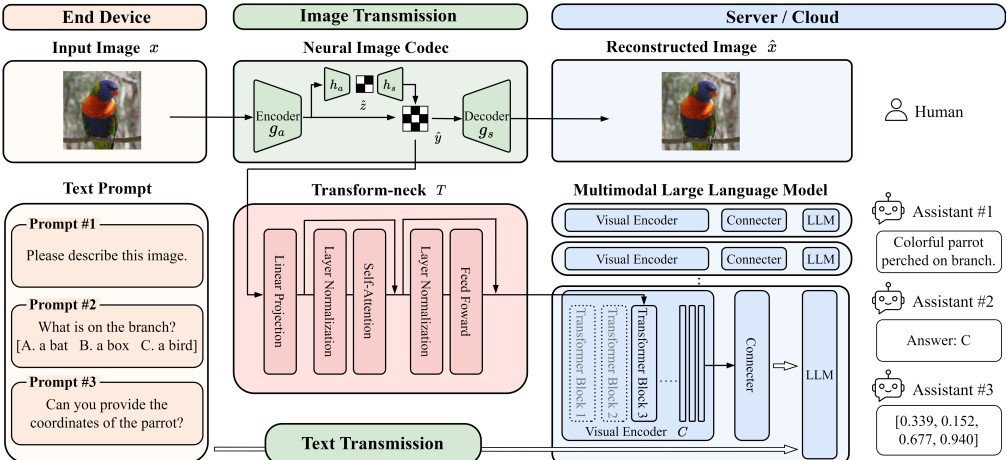

Figure 2: Overall architecture of the proposed method.

features of recognition networks, while (Feng et al., 2022) learn the omnipotent features suitable for various tasks in a self-supervised manner and fine-tune each task network tail on such features.

It is crucial to note that none of the coding for machine methods considers MLLMs at the receiver side. All the above-mentioned methods leverage back-propagation through recognition models to update the system or even re-train the recognition network itself, both of which are prohibitively expensive for MLLMs due to their huge scale. Therefore, the direct application of the same methods on MLLMs is almost infeasible. In addition, mainstream image coding for machines methods (e.g. Chen et al. (2023b); Ascenso et al. (2023)) remain mostly task-specific. They typically adopt a task-based loss, which restricts the resulting models to be optimized for a single task and recognition model, thus requiring re-training for each new task and incurring additional costs. We aim to be the first to propose a neural image compression system designed for MLLMs, achieved through a lightweight transform-neck and a surrogate loss, which bypasses the necessity of involving the entire billion-scale MLLM in the training process. Moreover, our surrogate loss incorporates a cross-entropy loss to bridge visual features with the text domain for MLLMs, complementing the feature-constraining distillation loss. This combination further differentiates our approach from existing methods.

# 3 PROPOSED METHOD

## 3.1 PRELIMINARIES: NEURAL IMAGE CODECS

The high-level architecture of a neural image codec is depicted in the top central green box in Figure 2. In a typical hyperprior-based neural image codec (Ballé et al., 2018), the key components include the main encoder $g_a$, the main decoder $g_s$, as well as the hyperprior encoder $h_a$ and decoder $h_s$. Given an RGB image $x \in \mathbb{R}^{3 \times H \times W}$, where $H$ and $W$ represent the height and width of the image, respectively, $g_a$ performs the analysis transform of $x$ and generates the image latent representation $y \in \mathbb{R}^{N \times \frac{H}{16} \times \frac{W}{16}}$, with $N$ indicating the channel size. To transmit $y$ more efficiently, it is first uniformly quantized into $\hat{y}$ and then entropy coded considering a learned prior distribution $p(\hat{y})$. This learned distribution is content dependent, thanks to the hyperprior encoder $h_a$ and decoder $h_s$. In particular, $h_a$ takes $y$ as input and produces the side information $z \in \mathbb{R}^{N_h \times \frac{H}{64} \times \frac{W}{64}}$, that is used to generate the learned distribution for entropy coding, where $N_h$ is the channel size of the side information. The quantized version of $z$, denoted as $\hat{z}$, is transmitted into the bitstream, in order to recover $\hat{y}$. Lastly, $\hat{y}$ undergoes the synthesis transform with $g_s$, which reconstructs the image $\hat{x} \in \mathbb{R}^{3 \times H \times W}$.

## 3.2 OVERALL FRAMEWORK

In this work, we focus on the scenario where MLLMs are hosted on the server side, while users on end devices need to perform inference on the remote model using both text and images as inputs. Given the necessity of incorporating image compression to ensure efficient transmission, we propose the first compression framework with the consideration of MLLMs as downstream application networks, aiming to mitigate the potential task performance drop caused by image compression.

Figure 2 illustrates our overall framework, which includes three major components: the neural image codec, our proposed transform-neck, and the MLLM. The depicted MLLM system adheres to a typical structure, consisting of a visual encoder, an LLM, and a connector component facilitating the transformation of features from the visual encoder to the LLM. Note that all the MLLMs are adopted off-the-shelf and without any update.

During inference, the input image at the end device is passed through an encoder $g_a$ to generate the quantized latents $\hat{y}$ for transmission. Next, $\hat{y}$ is directly passed through a lightweight transform-neck $T$ for transformation into a middle layer of the visual encoder of an MLLM. We opt to adapt the image latents rather than the reconstructed images because the image latents inherently contain the information needed for reconstructing the image, and potentially the semantic information for the downstream tasks (when the image encoder is guided properly). By skipping the image decoding process, our method offers reduced computational complexity while maintaining the task performance. The rest of the MLLM system operates without any changes to generate the desired output response.

In training, to address the challenge of back-propagating the task loss through the entire MLLM, we propose a novel surrogate loss that updates the system by back-propagating solely from the visual encoder (which is not re-trained), bypassing the billion-parameter LLM. In our work, we examine three distinct settings denoted as (d1), (d2) and (d3), as illustrated in Figure 1. Firstly, in (d1), we consider the practical scenario where a fixed off-the-shelf image codec pre-trained for human perception is directly used alongside our transform-neck. In this setting, our framework offers the option for users to decode the image latents $\hat{y}$ for reconstruction by using the decoder $g_s$ instead of the transform-neck. In this way, the quality of the decoded image is not affected, as the image codec is not updated in the present case. Then, we extend the analysis to scenarios (d2) and (d3) to examine the impact of jointly training the image codec and transform-neck. In (d2), the entire image codec undergoes re-training to accommodate both human and machine perception, while in (d3), the encoder is re-trained specifically for machine perception.

## 3.3 TRANSFORM-NECK

Our transform-neck is designed to be a lightweight module, consisting only of a linear projection, a self-attention mechanism, a feed-forward layer, and two layer norms, as shown in the central red box in Figure 2. Its purpose is to adapt the compressed image latents $\hat{y}$ into an efficient representation for consumption by the downstream MLLMs. In fact, rather than reconstructing the image and using it as input to the MLLM, we propose leveraging the latent representation directly.

Since the image encoder $g_a$ already functions as a feature extractor, similar to the early layers of the visual encoder $C$, we bridge the output of our transform-neck directly into the intermediate features of $C$, effectively integrating the image codec with the MLLM system. The decision on which initial layers to bypass depends on the specific type of visual encoder used in the MLLM. For instance, when using the CLIP visual encoder as $C$, we found that connecting the transform-neck to the third Transformer layer, bypassing the first two, yields optimal results (see Section 4.5 for ablation experiments justifying this design). Note that skipping the initial layers of $C$ further reduces computational complexity of our framework. We denote the partial visual encoder as $C'$.

## 3.4 SURROGATE LOSS

To avoid involving huge MLLMs in the training process, and thus bypassing back-propagation through their entire structure, we propose a surrogate loss $\mathcal{L}_S$ which is back-propagated through only the partial visual encoder $C'$. Specifically, we design our surrogate loss to consist of two terms: distillation loss $\mathcal{L}_{dist}$ and cross-entropy loss $\mathcal{L}_{CE}$.

To retain the downstream MLLM performance, the resulting features $C'(T(\hat{y}))$ when using our transformed latents should resemble closely those obtained when inputting the uncompressed image into $C$, that is $C(x)$. To this end, we introduce the following distillation loss $\mathcal{L}_{dist}$, aiming to minimize the Mean Squared Error (MSE) between the two output features:

$$\mathcal{L}_{dist} = \text{MSE}(C'(T(\hat{y})), C(x)). \tag{1}$$

In addition to the distillation loss, we propose incorporating a second term, which is the cross-entropy loss $\mathcal{L}_{CE}$ calculated using the following procedure. We take a classification dataset consisting of

Table 1: Application scenarios for our method with corresponding training objective.

| Application Scenario | Update Image Codec | Human Viewing | Phase 1 Loss Function | Phase 2 Loss Function |
|---|---|---|---|---|
| (d1) Human perception | ✗ | ✓ | $\mathcal{L}_S$ | — |
| (d2) Multi-task | ✓ | ✓ | $\mathcal{L}_S$ | $R + \lambda(\gamma d_{recon}(x, \hat{x}) + \delta\mathcal{L}_{dist})$ |
| (d3) Machine perception | ✓ | ✗ | $\mathcal{L}_S$ | $R + \lambda\mathcal{L}_{dist}$ |

$n$ images and $m$ text labels $\{l_1, l_2, \ldots, l_m\}$, which correspond to $m$ distinct classes. Each image belongs to one of these $m$ classes. Next, we compute: (1) the $n$ image embedding $C'(T(\hat{y}))$ and (2) the $m$ text embeddings $C_t(l_j)$, where $C_t$ is a text encoder applied to each text label $l_j$. In particular, we use the CLIP text encoder, independently of the visual encoder integrated into the MLLM under consideration. For each image, we then calculate the cosine similarity $\text{CS}(\cdot, \cdot)$ between its image embedding $C'(T(\hat{y}))$ and each of the $m$ text embeddings $C_t(l_j)$ for $j = 1, 2, \ldots, m$. This produces a vector $v$ of similarity scores between the given image and all the $m$ classes, that is:

$$v = [\text{CS}(C'(T(\hat{y})), C_t(l_1)), \ldots, \text{CS}(C'(T(\hat{y})), C_t(l_m))]. \tag{2}$$

The resulting similarity vector is transformed into a probability distribution over the $m$ classes using a softmax function. Finally, the cross-entropy loss is computed with respect to the corresponding one-hot encoded ground truth label vector $\mathbf{t}$, thus:

$$\mathcal{L}_{CE} = \text{CE}(\text{Softmax}(v), \mathbf{t}). \tag{3}$$

This approach aims to bridge the visual features to the text domain specifically for MLLM-based vision tasks, distinguishing our method from existing works in the field of coding for machines. We validate the effectiveness of our loss function design in the ablation study (Section 4.5).

## 3.5 TRAINING PROCEDURE

To explore the capabilities of our method under the application scenarios introduced in Section 3.2, we employ a two-phase training procedure that adapts to different scenarios. The first phase is shared for all scenarios and focuses on training the transform-neck exclusively. The second phase, applicable only to scenarios where the codec can be updated, involves joint optimization of both the transform-neck and the image codec. Table 1 provides a summary of the training procedure.

**Phase 1: Transform-neck Training** In the first phase, we train only the transform-neck using a progressive training strategy with the surrogate loss $\mathcal{L}_S$ and an off-the-shelf image codec. This phase is divided into three stages:

$$\mathcal{L}_S = \begin{cases} \mathcal{L}_{CE}, & \text{epoch} < E_1, \\ \alpha\mathcal{L}_{CE} + \beta\mathcal{L}_{dist}, & E_1 \le \text{epoch} < E_2, \\ \mathcal{L}_{dist}, & \text{epoch} \ge E_2, \end{cases} \tag{4}$$

where the weighting factors $\alpha$ and $\beta$ are set with a ratio of 1:100 for the two loss terms, and $E_1, E_2$ are empirically set to 20 and 40, respectively, in our experiments. This initial phase ensures that the transform-neck learns the transformation to align with the target latent space.

**Phase 2: Joint Optimization** For (d2) and (d3), where the image codec is allowed to be re-trained to produce latent representation more suitable for machine perception, the second phase is introduced with transform-neck and image codec jointly updated after phase 1 converges.

The scenario (d2), referred to as multi-task, aims to accommodate both human and machine perception. As a result, it is trained jointly with the transform-neck on both the distillation loss and traditional rate-distortion loss, i.e. $\mathcal{L}_{d2} = R + \lambda(\gamma d_{recon}(x, \hat{x}) + \delta\mathcal{L}_{dist})$, where $R = -\log p(\hat{z}) - \log p(\hat{y}|\hat{z})$ is the estimated rate of $\hat{y}$ and $\hat{z}$, and $d_{recon}$ is the reconstruction loss calculated as the MSE between the uncompressed image $x$ and the reconstructed image $\hat{x}$. The hyper-parameter $\lambda$ controls the rate-distortion trade-off, while $\gamma$ and $\delta$ weight the two losses.

In (d3), where the downstream applications do not require image reconstruction, the encoder and transform-neck are jointly optimized to minimize the trade-off cost between the rate $R$ and the distillation loss $\mathcal{L}_{dist}$, thus disregarding the reconstruction quality. The training objective is thus $\mathcal{L}_{d3} = R + \lambda\mathcal{L}_{dist}$.

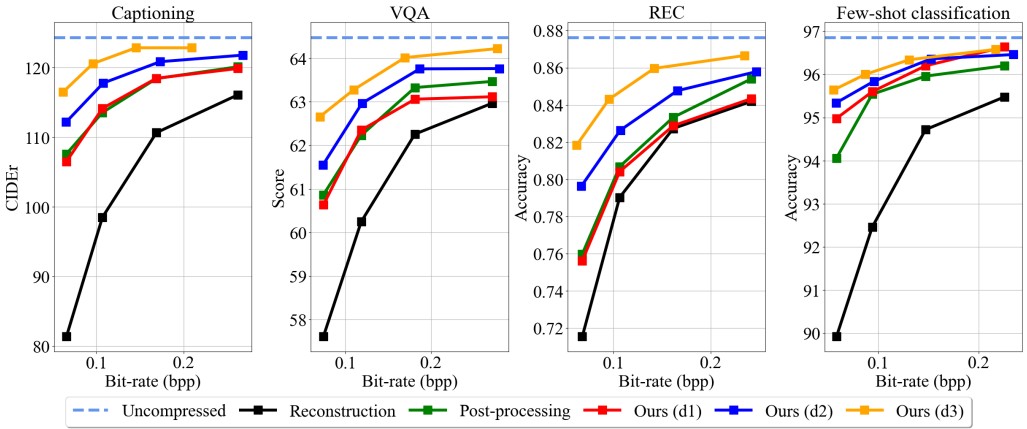

Figure 3: Rate-accuracy comparison using various MLLMs on several tasks.

Table 2: Evaluated tasks with corresponding dataset and MLLM.

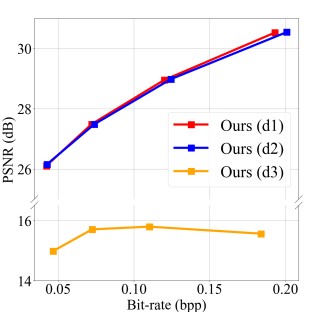

| Task | Dataset | MLLM |
|------|---------|------|
| Captioning | COCO Karpathy Test (Karpathy & Fei-Fei, 2015) | LLaMA-Adapter (Zhang et al., 2024a) |
| VQA | SEED-Bench (Li et al., 2023a) | Honeybee (Cha et al., 2024) |
| REC | RefCOCO-val (Kazemzadeh et al., 2014) | Shikra (Chen et al., 2023a) |
| Few-shot classification | ImageNet (Deng et al., 2009) | V2L-Tokenizer (Zhu et al., 2024b) |

Figure 4: Reconstruction performance comparison on Kodak.

## 4 EXPERIMENTAL RESULTS

### 4.1 EXPERIMENTAL SETTING

**Training Details and Datasets.** We utilize ELIC (He et al., 2022) as our image codec, which outputs image and hyperprior latents with $N = 320$ and $N_h = 192$, respectively. ELIC is trained for human perception and adheres to the training strategy outlined in He et al. (2022), using 8,000 images of the highest spatial resolution selected from ImageNet dataset. Four models are trained for four different rate points, corresponding to $\lambda = [0.004, 0.008, 0.016, 0.032]$ in (He et al., 2022). For each of our scenarios (d1), (d2) and (d3), separate transform-necks are trained on ImageNet dataset (Deng et al., 2009) for individual $\lambda$ values. For the scenario (d2) specifically, we find empirically that fixing the ratio $\gamma : \delta = 60 : 1$ leads to a good trade-off between human and machine perception. Given that most MLLMs adopt the pre-trained visual encoder of CLIP ViT-L/14 (Radford et al., 2021) for image modality, as discussed in Section 2.1, we use the CLIP visual encoder as $C$ for training and conduct our primary experiments on MLLMs that incorporate it. It is worth noting that, since we consider MLLMs sharing the same visual encoder, we do not need to train separate systems for the different MLLMs or tasks. Furthermore, we provide additional experiments in Section 4.6 with MLLMs that use visual encoders other than CLIP ViT to demonstrate the generalizability of our approach.

**Targeted MLLM-based Vision Tasks.** To validate the generalization ability of our proposed method, we evaluate its performance on four different MLLM systems for four different tasks. The tasks, datasets, corresponding MLLMs, and metrics are listed in Table 2. These configurations follow the settings outlined in their original papers and the accompanying code, except for the few-shot classification task due to the inaccessibility of the code. We thus design a 5-way 1-shot classification scenario to evaluate the performance with in-context learning; the detailed setting is described in supplementary material. All MLLMs are used off-the-shelf without any fine-tuning.

**Baselines.** We introduce two baseline methods for comparison. The first one, denoted as *Reconstruction*, involves inputting the reconstructed image generated by ELIC to the MLLM system.

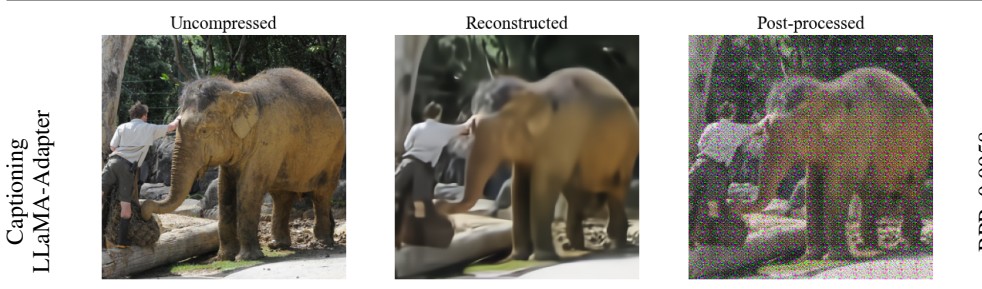

Figure 5: Visualization examples of our proposed method in (d1), *Reconstruction*, and *Post-processing* on image captioning with LLaMA-Adapter and REC with Shikra.

The second one, denoted as *Post-processing*, adapts the reconstructed image to MLLMs through a U-Net (Ronneberger et al., 2015) post-processing network, which is trained using the same surrogate loss as that adopted by our method. We remark that these image-domain baselines incur higher complexity than our lightweight transform-neck, as they involve decoding the image and potentially processing it further with the post-processing network.

## 4.2 PERFORMANCE COMPARISON

Figure 3 illustrates the performance of the baseline methods and our proposed scheme for the three examined scenarios with regards to two aspects: coding bit-rate, calculated as bits per pixel (bpp), and task performance. When comparing the baselines and our method in scenario (d1), where the original ELIC is trained solely for human perception, we make the following observations. (1) Straightforwardly using the reconstructed images generated by a codec trained for human perception leads to a significant performance drop across all the tasks (*Reconstruction*). Such performance decline is expected because the MLLMs are not trained with compressed images, thus hindering their recognition performance. This highlights the necessity of adapting image compression and/or image latents to MLLMs. (2) In contrast, our transform-neck method successfully boosts the performance using the same latent representations for reconstructing the image in *Reconstruction*, confirming the effectiveness of the proposed latent transformation without the decoding process. (3) *Post-processing* is able to reach comparable performance to our (d1), offering another viable solution to the problem. However, it is worth noting that *Post-processing* requires relatively higher computational complexity with respect to our transform-neck method, rendering our approach preferable (see Section 4.4).

Next, we evaluate the effects of allowing the image codec to be re-trained. First, we observe that (d2) outperforms both (d1) and *Post-processing*. This indicates that fine-tuning the encoder indeed results in a more suitable latent representation that can be better adapted to MLLMs. When examining the extreme setting (d3), we see significant further improvement in the task performance, approaching the performance upper bound with uncompressed images. This improvement comes at the cost of the image reconstruction quality, which, however, is not relevant in (d3). Figure 4 illustrates the rate-visual quality curves associated with the three scenarios. Interestingly, (d2) exhibits only a marginal PSNR drop compared to (d1), while (d3) significantly compromises the quality of the decoded image. We stress that our framework (i.e. the surrogate loss and transform-neck) is able to accommodate different application scenarios, allowing for a variable trade-off between the task performance and the image reconstruction quality.

Table 3: Comparison of the kMACs/pixel and model size. The table omits the shared components of the two methods, i.e. the image encoder, the partial CLIP visual encoder, the connector, and the LLM.

| Method | Component | Params (M) | | kMAC/pixel | |
|---|---|---|---|---|---|
| **Ours (d1, d2, or d3)** | Transform-neck | 13.19 | | 52.795 | |
| *Post-processing* | Decoder
Post-processing network
First 2 layers of CLIP visual encoder | 7.34
31.04
25.78 | 64.16
**(+386%)** | 112.00
835.72
70.24 | 1017.96
**(+1828%)** |

(a) Partial CLIP visual encoder    (b) Training objectives    (c) Different image codecs

Figure 6: Rate-accuracy comparison for three ablation studies evaluated with image captioning task.

## 4.3 VISUALIZATION OF THE RESULTS

We present the visualization of outcomes with downstream MLLM-based vision tasks in Figure 5. Our method (d1) is compared with the two baseline methods, *Reconstruction* and *Post-processing*, with particular focus on how these models work at low bitrates to reflect a bandwidth-limited scenario. In the second and third columns, we visualize the reconstructed and post-processed images from the two baselines, respectively, which exhibit drastically different characteristics. The former (*Reconstruction*) produces blurry and smooth images, while the latter (*Post-processing*) introduces some artificial patterns into the post-processed images. Compared with the baselines, our method yields better MLLM results. More visualization are presented in Section A.3 of the supplementary material.

## 4.4 COMPLEXITY ANALYSIS

Table 3 compares the computational complexity between our proposed method and *Post-processing* baseline in terms of model size and the kilo-multiply-accumulate-operations per pixel (kMACs/pixel). Note that our method in Table 3 refers to any of (d1), (d2), and (d3), since they share the same computational complexity characteristics at inference time. Our method offers a lightweight solution with only 13 million parameters, as opposed to 64 million parameters with the post-processing approach. Moreover, in terms of kMAC/pixel, the difference stands out even more, considering that the post-processing network operates at the full image resolution while our method operates in the latent domain, where the image latents have a much smaller spatial resolution.

## 4.5 ABLATION STUDIES

The following ablation experiments are performed based on (d1) to justify our design choices.

**Training Objective.** Figure 6 (b) presents the performance of our method when trained exclusively with the cross-entropy loss or distillation loss. It is observed that training with only the cross-entropy loss results in a significant performance drop. Although providing a good initial update direction, this strategy is unable to learn an effective transformation for MLLMs. Instead, training solely with the distillation loss fails to update the transform-neck properly and leads to far inferior performance. This is potentially due to the stringent requirement of fitting the exact feature representations. Our proposed method, not a simple application of distillation, achieves the highest performance.

To further support this finding, we present the following analysis in Figure 7: we first calculate the mean squared error (MSE) between CLIP visual encoder output tokens derived from uncompressed

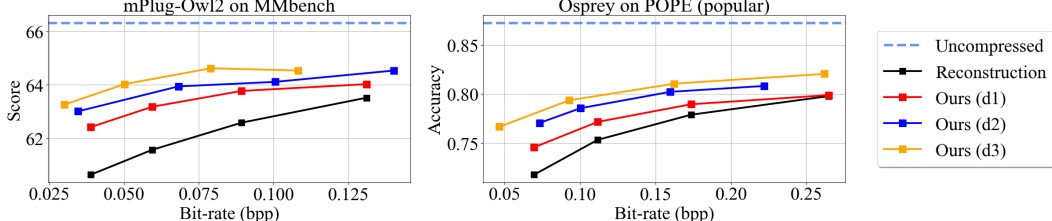

Figure 7: Visualization of MSE reduction on CLIP visual encoder output tokens before and after training using different loss functions. Darker red colors indicate greater MSE reduction.

Figure 8: Rate-accuracy comparison on two MLLMs not utilizing CLIP ViT visual encoder: mPLUG-Owl2 (Ye et al., 2024) on MMBench and Osprey (Yuan et al., 2024) on POPE (popular setting).

images and from our transform-neck, both before and after the transform-neck has been trained. Then, we compute the difference between these MSEs to measure the improvement achieved by the specific training objectives all with equal training steps. Figure 7 shows that the cross-entropy loss reduces feature matching errors primarily in foreground object regions, while the distillation loss reduces global matching errors. Our proposed progressive training strategy integrates these two losses, leading to a greater reduction in MSE and thus improved rate-accuracy performance.

**Partial CLIP Visual Encoder.**    This experiment investigates the proper number of Transformer layers to remove from the CLIP visual encoder in order to strike a good balance between complexity and performance. As shown in Figure 6 (a), removing the first one or two layers achieves similar performance, whereas removing four or eight layers results in a noticeable performance drop. We thus remove the first two layers.

**Different Image Codecs.**    Figure 6 (c) presents the performance comparison between our method and *Reconstruction* when they are tested with ELIC and TIC (Lu et al., 2022a;b). TIC is a Transformer-based codec, whereas ELIC is a convolutional neural network-based codec. We see that our transform-neck still outperforms *Reconstruction* by a significant margin when the image codec is changed from ELIC to TIC. This indicates that our method is still effective on a different type of image codec.

## 4.6    GENERALIZATION

While we utilize the CLIP ViT visual encoder in our main experiments due to its wide popularity, our proposed method is applicable to various downstream MLLMs regardless of the visual encoder they adopt. As illustrative examples, Figure 8 presents the rate-accuracy performance of our re-trained scheme applied to two MLLMs that do not use the pre-trained CLIP ViT visual encoder: (1) mPlug-Owl2 (Ye et al., 2024) with a custom-trained ViT visual encoder, and (2) Osprey (Yuan et al., 2024) with a CLIP ConvNeXt-based visual encoder. Our method under all three settings clearly outperforms the *Reconstruction* baseline, confirming the generalizability of the proposed framework.

## 5    CONCLUSION

This paper proposes the first image compression system tailored to Multimodal Large Language Models (MLLMs). It introduces a transform-neck that bridges the compressed image latents and the intermediate layer of the visual encoder. By using our proposed surrogate loss, we avoid involving the entire MLLM in the training process and ensure downstream task performance. With lower computational complexity, our method has demonstrated effectiveness across a wide variety of tasks, MLLMs, and neural image codecs, outperforming other baselines in extensive experiments. One consideration is that this paper focuses solely on the image compression aspect, leaving the exploration of MLLM-based video or audio coding for future work.

ACKNOWLEDGMENTS

This work is supported by National Science and Technology Council, Taiwan under the Grant NSTC 113-2634-F-A49-007-, MediaTek, and National Center for High-performance Computing, Taiwan.

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

# A SUPPLEMENTARY MATERIAL

## A.1 IMPLEMENTATION DETAILS

**Training.** We use the Adam optimizer, configured with $\beta_1$ at 0.9, $\beta_2$ at 0.999, $\epsilon$ at $10^{-8}$. Weight decay is disabled. The transform-neck for each rate point undergoes training on an RTX 4090 for approximately three days during the training stage.

**Evaluation.** For few-shot classification with V2L-Tokenizer (Zhu et al., 2024b), we design a 5-way 1-shot classification evaluation scenario. In particular, we generate 5000 groups of images from ImageNet dataset, where each group consists of five randomly sampled images from different classes, serving as the sample images, and one new image from one of the classes as the query image.

Different MLLM is utilized for the evaluation of our proposed method on each task. In Table 4, we provide the detailed specifications of the MLLM used in our evaluation.

Table 4: The specifications of the MLLM used in our tasks.

| Task | Model | LLM |
|---|---|---|
| Captioning | LLaMA-Adapter v1 (Zhang et al., 2024a) | LLaMA-7B (Touvron et al., 2023a) |
| VQA | Honeybee-C-7B-M144 (Cha et al., 2024) | Vicuna-7B (Chiang et al., 2023) |
| REC | Shikra-7B (Chen et al., 2023a) | LLaMA-7B (Touvron et al., 2023a) |
| Few-shot classification | V2L-Tokenizer (Zhu et al., 2024b) | LLaMA2-7B (Touvron et al., 2023b) |

## A.2 COMPARISON WITH VVC

Figure 9 compares *Reconstruction* and our method in (d1) using ELIC, with the state-of-the-art traditional codec VVC (VTM 17.0 intra coding). We set the QPs to $[37, 40, 43, 46, 49]$ for VVC. It is observed that VVC performs worse than *Reconstruction* across all the tasks, which is potentially due to (1) the small spatial resolution (256x256) of input images that is not optimal for VVC, (2) its inferior rate-distortion performance compared to ELIC as reported in (He et al., 2022).

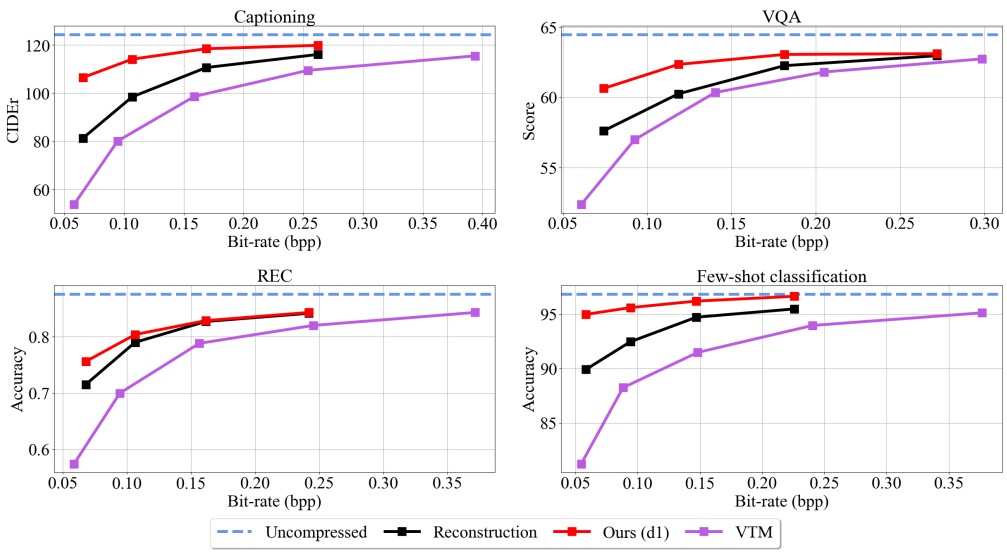

Figure 9: Rate-accuracy comparison using VTM on several tasks.

### A.3 MORE VISUALIZATION

We present additional visualization results on four different evaluation tasks, including image captioning (Figure 12), visual question answering (VQA) (Figure 13), referring expression comprehension (REC) (Figure 14), and few-shot classification (Figure 15).

### A.4 LICENSE OF ASSETS USED

Table 5 summarizes the used assets in our work along with their license terms.

Table 5: List of assets used in the paper with their corresponding license.

| Assets | Licenses |
|---|---|
| ImageNet Deng et al. (2009) | Custom license. Available at https://image-net.org/download.php |
| COCO Lin et al. (2014) | CC BY 4.0 |
| SEED-Bench Li et al. (2023a) | Apache 2.0 |
| LLaMA-Adapter Zhang et al. (2024a) | GPL-3.0 |
| Honeybee Cha et al. (2024) | Source code: Apache 2.0
Pretrained weights: CC BY-NC 4.0 |
| Shikra Chen et al. (2023a) | CC BY-NC 4.0 |
| V2L-Tokenizer Zhu et al. (2024b) | No license provided.
Code available at https://github.com/zh460045050/V2L-Tokenizer |

### A.5 ADDITIONAL PERFORMANCE COMPARISON

Figure 10 presents an additional performance comparison of our proposed method on the same task (POPE benchmark (Li et al., 2023c)) with two different MLLMs, Honeybee (Cha et al., 2024) and Shikra (Chen et al., 2023a).

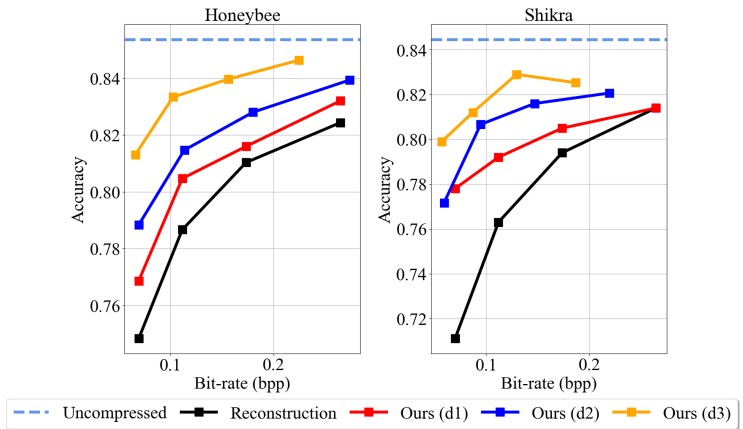

Figure 10: Performance comparison on POPE benchmark (Li et al., 2023c) with different MLLMs.

### A.6 COMPARISON WITH TOKEN REDUCTION METHODS

Token reduction methods (Shi et al., 2024; Li et al., 2024) aim to reduce the number of visual tokens for lowering the inference computational cost of MLLMs. These methods differ fundamentally from our method as they do not consider the transmission of visual tokens in compressed form. In contrast, our work encodes and transmits the image in compressed form, where the compressed image latents are adapted to suit the downstream MLLM and image reconstruction tasks. The focus of our work is at maintaining downstream performance while reducing transmission cost and decoding inference cost. Figure 11 illustrates the different costs associated with each component in the system.

Ideally, one might propose generating visual tokens on the end device and using token reduction techniques as a compression method to reduce transmission bandwidth. However, this approach

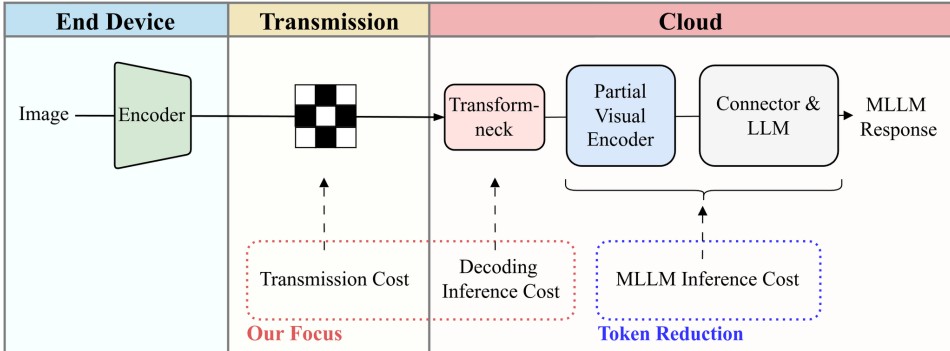

Figure 11: High-level architecture of our proposed method and different cost associated with each component.

would impose significant computational demands on the end device, making it impractical for coding for machines scenarios, where the primary goal is to offload heavy feature extraction computations to the cloud.

Notably, our method and the token reduction technique could potentially complement each other to develop a more efficient system. For instance, our approach allows to save transmission bandwidth and reduce cloud-side complexity by eliminating the need for image decoding, then token reduction techniques can further optimize complexity on the cloud.

Task: Captioning
Model: LLaMA-Adapter

| Uncompressed | Reconstructed | Post-processed |
|---|---|---|
| 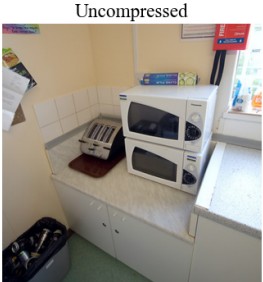 | 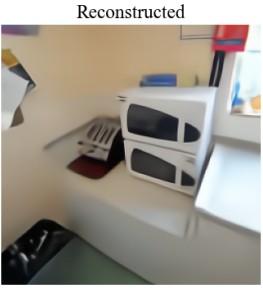 | 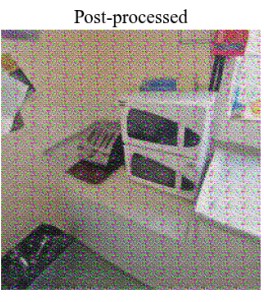 |

BPP: 0.0725

*Reconstruction*: A microwave and a computer sitting on a desk.
*Post-processing*: A microwave and a refrigerator sitting on top of a table.
Ours (d1): A microwave and a toaster oven on a counter.

| Uncompressed | Reconstructed | Post-processed |
|---|---|---|
| 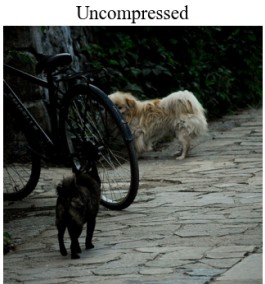 | 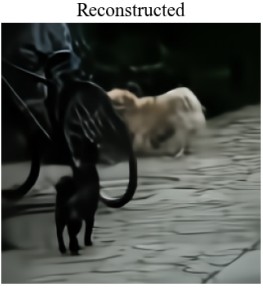 | 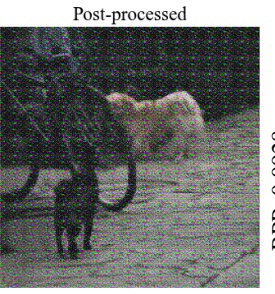 |

BPP: 0.0928

*Reconstruction*: Two cats are standing on the ground near a bench.
*Post-processing*: A dog and a cat are standing on a sidewalk.
Ours (d1): Two dogs are standing near a bicycle on a sidewalk.

| Uncompressed | Reconstructed | Post-processed |
|---|---|---|
| 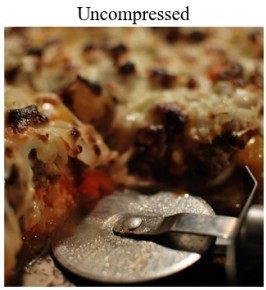 | 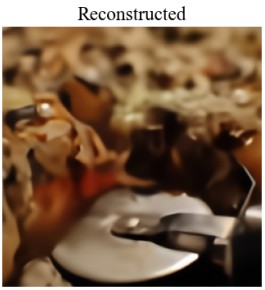 | 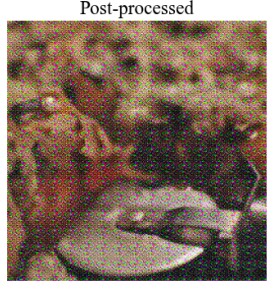 |

BPP: 0.0910

*Reconstruction*: A blurry picture of a blender with a knife.
*Post-processing*: A close up of a blurry image of a bug.
Ours (d1): A close up of a knife cutting into a pizza.

| Uncompressed | Reconstructed | Post-processed |
|---|---|---|
| 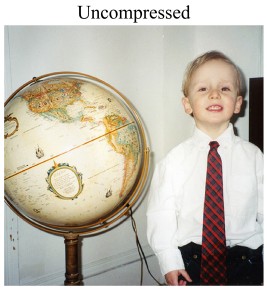 | 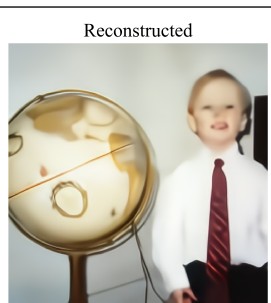 | 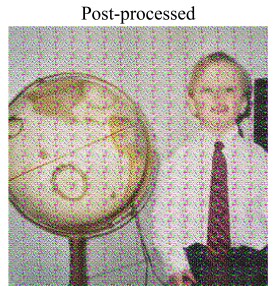 |

BPP: 0.0899

*Reconstruction*: A young boy in a red shirt and tie posing for a picture.
*Post-processing*: A young boy standing in front of a wall with a clock.
Ours (d1): A young boy in a tie and a white shirt.

Figure 12: Visualization examples of our proposed method in (d1), *Reconstruction*, and *Post-processing* on image captioning with LLaMA-Adapter.

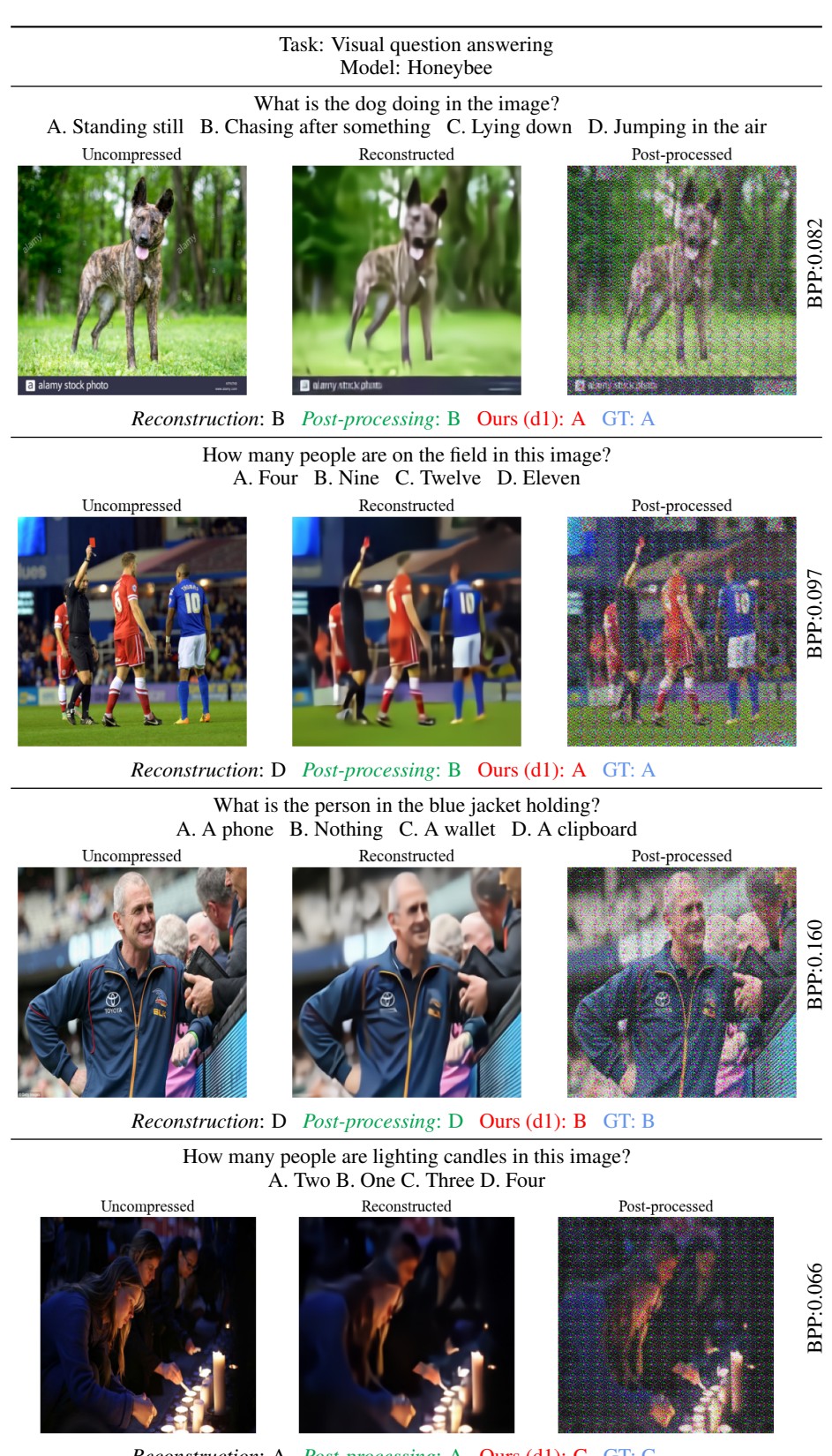

Figure 13: Visualization examples of our proposed method in (d1), *Reconstruction*, and *Post-processing* on VQA with Honeybee.

Task: Referring expression comprehension
Model: Shikra

Guide me to the location of brown bear within the image  by providing its coordinates.

Point me to the location of wine glass far left in the picture  by providing its coordinates.

Can you assist me in locating right female cop in , and then provide its coordinates?

In the photograph , could you pinpoint the location of
person holding a snowboard and tell me its coordinates?

Figure 14: Visualization examples of our proposed method in (d1), *Reconstruction*, and *Post-processing* on REC with Shikra.

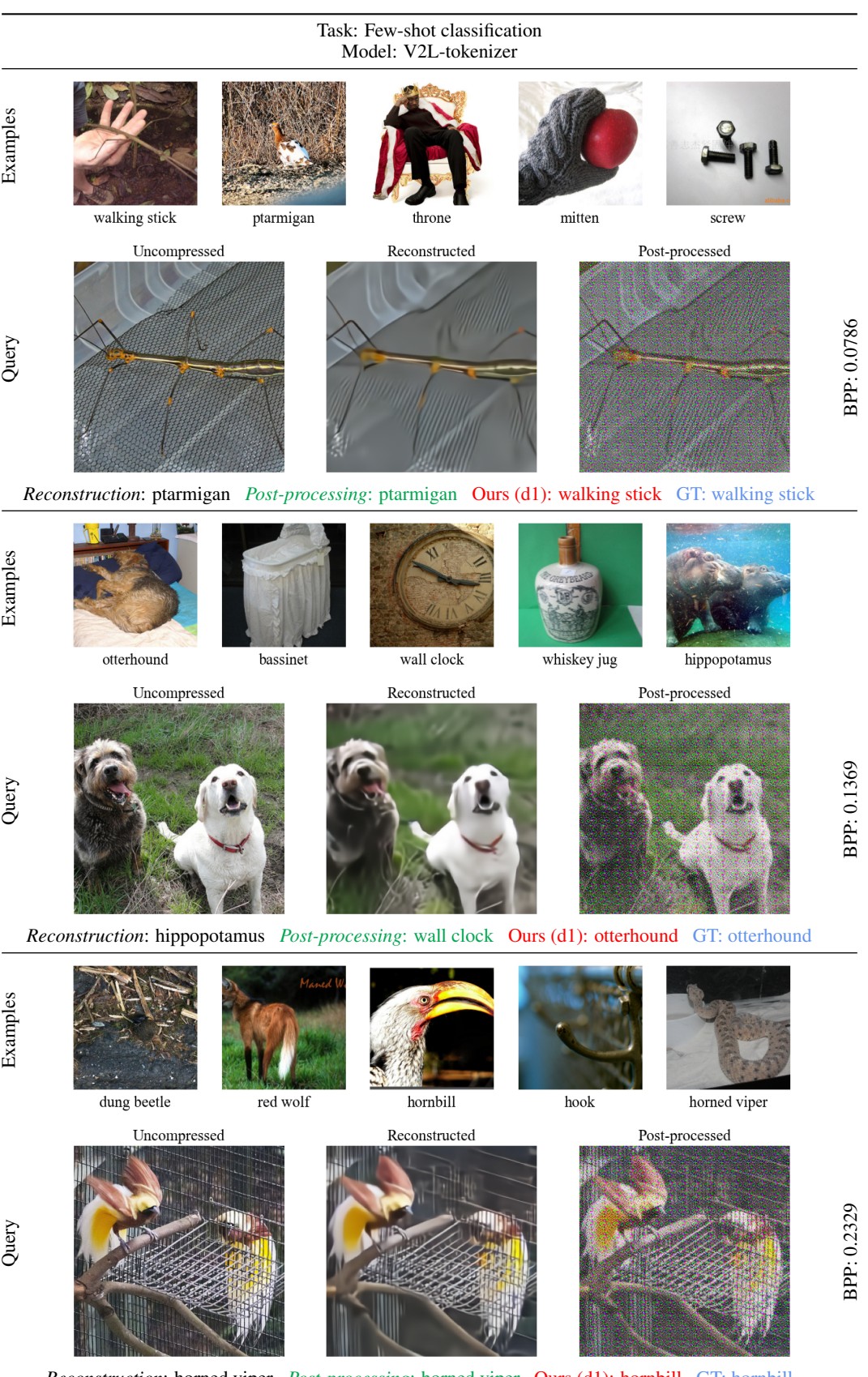

Figure 15: Visualization examples of our proposed method in (d1), *Reconstruction*, and *Post-processing* on few-shot classification with V2L-tokenizer.

