# OpenReview forum: "Bridging Compressed Image Latents and Multimodal Large Language Models"
_ICLR.cc/2025/Conference — ICLR 2025 Poster_

### Official Review · Reviewer_HmZL · 2024-10-26

**Soundness:** 2
**Presentation:** 2
**Contribution:** 2
**Rating:** 6
**Confidence:** 2

**Summary:**

This paper explores methods to compress and decode images such that the decoded images can be successfully used with Large MLLMs, addressing the substantial overhead required for transmitting images to cloud-hosted MLLMs. Existing image encoding works for machine vision face a common problem: MLLM models are too large to train the neural image codec. To address the computational overhead of image compression and encoding in the context of MLLMs, this paper proposes a lightweight transform-neck and a novel surrogate loss. The lightweight transform-neck aims to align the features of the compressed images with the outputs of the visual encoder, avoiding the need for fully image decoding. The proposed surrogate loss allows the neural image codec to be trained using only the vision encoder, avoiding the need for backpropagating through the entire MLLM. The proposed method can be applied to different MLLMs and various neural image codecs in various application scenarios.

**Strengths:**

1. This paper claims to be the first work specifically targeting neural image coding for MLLMs.
2. It introduces a lightweight transform-neck that aligns the encoded image latents with the MLLM visual encoder outputs, thereby avoiding the decode operation in neural image coding and reducing computational complexity.
3. The proposed surrogate loss allows updating the neural image coding model via the Visual Encoder without backpropagating through the entire MLLM, hence reducing the computation required.
4. The proposed method is applicable to different downstream MLLMs, applications, and neural image codecs.

**Weaknesses:**

1. The motivation of this paper—studying image compression under MLLM to avoid transmission overhead—seems to lack practical significance. The primary goal of image compression is to reduce image resolution, which might be meaningless for MLLMs:
* First, most MLLMs [1] use CLIP Image Encoder, which limits image resolution to 224x224. This resolution does not incur significant transmission overhead in current network communications and is negligible compared to the inference time of MLLMs.
* Second, while some works [2] explore using high-resolution images as input, they usually crop high-resolution images into several low-resolution sub-images, often achieving better performance compared to compressed images. Therefore, researching how to compress images as MLLM input seems unnecessary.
* Lastly, the authors do not explain the advantages of using compressed images as MLLM input over exploring more compact Image Encoders [3].

2. The proposed method requires training different transform-necks and codecs for different downstream tasks, further leading me to question the necessity and rationale of this work.

Reference
* [1] Visual Instruction Tuning, NeurIPS 2023
* [2] About When do we not need larger vision models?, ECCV 2024
* [3] TokenPacker: Efficient Visual Projector for Multimodal LLM, arXiv 2024

**Questions:**

Please refer to the Weaknesses

---

> ### Author Response · Authors · 2024-11-20
> **Response to Reviewer HmZL (1/2)**
>
> > **[W1] The motivation of this paper seems to lack practical significance. The primary goal of image compression is to reduce image resolution. [...] This resolution (224x224) does not incur significant transmission overhead in current network communications.**
>
> We would like to respectfully clarify that the primary goal of image compression is NOT to reduce image resolution, but to reduce image storage or transmission costs while preserving essential information. Our approach is specifically designed to **adapt the compressed image latents to facilitate MLLM-based recognition tasks while minimizing the transmission cost**.
>
> It is also a misconception that transmitting images of size 224x224 does not incur significant transmission overhead. The reasons are two-fold:
> - The volume of images/videos for recognition tasks is growing explosively. Many end devices with limited resources send images to the cloud for recognition. Given this large-scale image/video transmission across various networks, **compression remains necessary to meet the bandwidth and energy consumption requirements** even for images of size 224x224, not to mention higher resolution images. These observations motivate the ongoing international standardization projects on image/video coding for machines in ISO/IEC MPEG `[R1]` and ISO/IEC \& ITU-T JPEG `[R2]`, underscoring the importance of image/video compression for machines whether they are traditional recognition networks or modern MLLMs.
>
> - Additionally, there is an increasing number of MLLMs exploring video-based applications, where videos are often sub-sampled temporally as a sequence of images. This trend also calls for efficient compression of images/videos for MLLM-based tasks.
>
>
> ---
>
> > **[W1] Some works explore using high-resolution images as input, they usually crop high-resolution images into several low-resolution sub-images.**
>
> We like to clarify that cropping and compression are two distinct mechanisms with different objectives. Simply cropping high-resolution images into low-resolution patches does not effectively reduce the transmission overhead. We stress that significant bandwidth is still required even for cropped sub-images if no efficient compression is applied.
>
> ---
>
> > **[W1] The authors do not explain the advantages of using compressed images as MLLM input over exploring more compact Image Encoders `[R3]`.**
>
> Token reduction methods (such as `[R3]`) differ fundamentally from our approach. They aim to reduce the number of visual tokens in MLLMs to lower computational complexity, and do NOT transmit visual tokens in compressed form.
> In contrast, our work addresses a fundamental preliminary step. Since the image must be transmitted to a cloud server for MLLM processing, we focus on **bridging the gap between compressed image latents and their suitability for downstream MLLM and image reconstruction tasks.**
>
> Ideally, one might propose generating visual tokens on the end device and using token reduction techniques as a compression method to reduce transmission bandwidth. However, this approach would impose significant computational demands on the end device, making it impractical for coding for machines scenarios, where the primary goal is to offload heavy feature extraction computations to the cloud.
>
> Notably, our method and the token reduction technique could potentially complement each other to develop a more efficient system. For instance, our approach allows to save transmission bandwidth and reduce cloud-side complexity by eliminating the need for image decoding, then token reduction techniques can further optimize complexity on the cloud.

---

> > ### Comment · Reviewer_HmZL · 2024-11-25
> >
> > > [W1] The motivation of this paper seems to lack practical significance. The primary goal of image compression is to reduce image resolution. [...] This resolution (224x224) does not incur significant transmission overhead in current network communications.
> >
> > I appreciate your clarification that the motivation of your work is to adapt the compressed image latents to facilitate MLLM-based recognition tasks while minimizing the transmission cost, rather than aiming to reduce image resolution. Unfortunately, as I am not an expert in this specific area, I am unable to fully assess the practical contribution of this approach. If you wish to further convince me, it would be helpful to provide concrete figures regarding the actual transmission cost of a 224x224 image, as well as the inference cost associated with the corresponding MLLMs. This would better highlight the trade-offs and the extent of the transmission savings.
> >
> > ---
> >
> > > [W1] Some works explore using high-resolution images as input, they usually crop high-resolution images into several low-resolution sub-images.
> > > [W1] The authors do not explain the advantages of using compressed images as MLLM input over exploring more compact Image Encoders [R3].
> >
> > Thank you for clarifying the difference between your approach and image crop as well as compact image encoders. Your method aims to reduce the cost during transmission, whereas image crop and compact image encoders aim to decrease the cost of model inference. However, these costs may not be equivalent. If the inference cost significantly outweighs the transmission cost, it might be possible to mitigate the transmission cost considerably using techniques such as parallel inference. Without concrete figures on the respective costs, I am unable to evaluate the practical rationale of your work.

---

> > > ### Author Response · Authors · 2024-11-29
> > > **Seeking your further feedback**
> > >
> > > Dear Reviewer HmZL, thank you for reviewing our rebuttal responses carefully. We do appreciate your effort in making sure that our paper is accessible to the readers of all familiarity levels. As per your invite ("If you wish to further convince me, it would be helpful to ..."), we have
> > > (1) provided concrete figures to address the transmission and inference costs,
> > > (2) explained why these two costs are independent, non-comparable factors in our context, and
> > > (3) updated our manuscript in response to your comments/suggestions.
> > > As the rebuttal period will end soon, we like to take this opportunity to seek your further feedback and look forward to your final favorable rating.

---

> > > > ### Comment · Reviewer_HmZL · 2024-11-30
> > > >
> > > > Thank you for your response. I fully understand the distinction between transmission overhead and model inference overhead, and I recognize that your work focuses on reducing transmission overhead.
> > > >
> > > > My primary concern is whether the model inference overhead significantly exceeds the transmission overhead. This issue is critical to determining the significance of this work in my view. If the model inference overhead is significantly greater than the transmission overhead, then reducing transmission overhead seems trivial for the inference of VLMs (Vision-Language Models).
> > > >
> > > > I would like to see specific numbers, such as the transmission overhead in seconds and inference overhead in seconds for an image of size 224x224.

---

> > > > > ### Author Response · Authors · 2024-12-01
> > > > > **Response to Reviewer HmZL**
> > > > >
> > > > > We thank you very much for your clarification and comment.
> > > > >
> > > > > Before providing the quantitative results, as requested, we would like to further explore the core of the issue.
> > > > > In our view, **comparing transmission and inference costs is not the central point.** Regardless of the inference cost, the value of compression is unquestionable in our scenario. Compression offers several benefits beyond reduced transmission time, like bandwidth saving, for example. These advantages make the use of compression systems for transmitting images from end devices to servers self-evident. We trust you share this perspective.
> > > > >
> > > > > Building on this, our work tackles a critical challenge: MLLMs, which are typically deployed on servers due to their computational demands, are usually trained on raw data, and their performance degrades significantly when using compressed images (as shown in Fig. 3 of the main paper). This is where our contribution lies. By introducing and training a bridge that connects the latent representation to the visual encoder, we achieve a substantial performance improvement with respect to working with compressed images. Please note that this performance improvement stands independently of the inference cost of the MLLM. Also, the advantages of having compressed the images remain intact, such as reducing transmission time, saving bandwidth, and more.
> > > > >
> > > > > With this premise in mind, and as per your request, we would like now to provide the numbers you requested. These numbers are estimated by assuming that the end user (1) takes a picture of size 224x224 and (2) uploads the image without any compression through a 5G network. On the cloud side, a machine equipped with A6000 is used and the task evaluated is the SEED Benchmark with Honeybee as the downstream MLLM.
> > > > >
> > > > > Notably, we measure the real-world and live “upload” transmission bandwidth of 5G networks in different areas, including Germany, Italy, and Taiwan. **[The attached screenshots](https://i.imgur.com/1Ac6XkO.png)** provide the upload bandwidth we have collected.
> > > > >
> > > > > According to these statistics, we could infer that the time needed to upload an image of size 224x224 is about 0.12 seconds if we use 10 Mbits/s as the average upload bandwidth. **Note that there are several cases where the actual upload bandwidth is lower than 10 Mbits/s, and thus the transmission time can even be higher. Also, it is envisioned that higher-resolution images may be used.**
> > > > >
> > > > > On the other hand, the average MLLM inference time per image is around 0.14 seconds (on A6000).  The average inference time per image is evaluated by measuring the inference time needed for inferencing all the images in the dataset divided by the total number of images. **Note that the inference time may be further reduced when performing on a more powerful GPU in the cloud.**
> > > > >
> > > > > In conclusion, the transmission time vs MLLM inference time is close to 1:1. With our proposed method, we effectively reduce the transmission time by 120x-240x, resulting in as low as 0.0005 seconds per image. That is, with compression, the transmission overhead in the present case is negligible.
> > > > >
> > > > > We hope our responses have addressed the concerns satisfactorily, and would be happy to address your additional comments (if any). We look forward to your final favorable rating.

---

> > > > > > ### Comment · Reviewer_HmZL · 2024-12-01
> > > > > >
> > > > > > Thanks to the author's further clarification and updated results, I have no additional concerns about this paper, so I will raise the score to Rating: 6: marginally above the acceptance threshold

---

> > > > > > > ### Author Response · Authors · 2024-12-01
> > > > > > > **Response to Reviewer HmZL**
> > > > > > >
> > > > > > > Thank you for your engagement during the rebuttal process, and we sincerely appreciate your effort and feedbacks.

---

> ### Author Response · Authors · 2024-11-20
> **Response to Reviewer HmZL (2/2)**
>
> > **[W2] The proposed method requires training different transform-necks and codecs for different downstream tasks.**
>
> We clarify that **our proposed method does NOT require re-training for every downstream task**. First, since one single MLLM is capable of performing multiple tasks `[R4]`, our trained system inherently supports multi-task inference. Second, our training framework, which back-propagates the surrogate loss at the output of visual encoder rather than that of the downstream MLLM, enables the resulting transform-neck and/or image codec to be **applied directly to EVEN DIFFERENT MLLMs** sharing the same visual encoder, WITHOUT the need for retraining. This feature further distinguishes our approach from many existing coding for machines methods that indeed require task-specific retraining.
>
> ---
>
> **We thank the reviewer for the insightful comments.** We hope our responses have addressed the concerns satisfactorily, and would greatly appreciate that the reviewer considers increasing the rating on this pioneering attempt to perform image coding for MLLM-based tasks.
>
> ---
>
> **References**
>
> `[R1]` "Use cases and requirements for Video Coding for Machines," ISO/IEC JTC 1/SC 29/WG 2 output document, N00190, 2022. \
> `[R2]` "JPEG AI Use Cases and Requirements," ISO/IEC JTC 1/SC 29/WG 1 output document, N100724, 2024.\
> `[R3]` W. Li et al., "TokenPacker: Efficient Visual Projector for Multimodal LLM", arXiv 2024. \
> `[R4]` H. Liu et al., "Visual instruction tuning", NeurIPS, 2024.

---

> > ### Comment · Reviewer_HmZL · 2024-11-25
> >
> > > [W2] The proposed method requires training different transform-necks and codecs for different downstream tasks.
> >
> > Thanks for clarifying this, I have no more questions on this problem. Hence, I will increase my score to **5: marginally below the acceptance threshold**

---

> > > ### Author Response · Authors · 2024-11-26
> > > **Response to Reviewer HmZL**
> > >
> > > Thank you for putting much effort to ensure that our paper is easily accessible to readers of all familiarity levels.
> > >
> > > Following your terminology, we like to clarify and distinguish between the transmission cost and inference cost, as **they are fundamentally two separate and non-comparable costs**. By the "transmission cost," we mean the number of bits required to send an image from an end device to the cloud, while the "inference cost" refers to the computational complexity (e.g. the number of the multiply–accumulate operations) needed to decode or adapt the image’s latents in a way suitable for the downstream MLLMs. These two costs are non-comparable. Our objective is NOT to trade-off between these two costs. Instead, **our framework reduces the transmission cost of sending an image from an end device to the cloud with a minimal impact on the recognition accuracy** of the downstream MLLM-based tasks. It also offers a feature that allows the image’s latents to be adapted directly to the downstream MLLMs without having to reconstruct the image. This helps save the “inference cost” as compared to a baseline that has to reconstruct and enhance the compressed image for the MLLM-based tasks. The **[attached figure](https://i.imgur.com/KPxHvhA.png)** illustrates different costs associated with each component. As per your request, we provide concrete figures as follows:
> > >
> > > * **[Transmission Cost]** Without compression, it requires 24 bits to transmit a single RGB pixel of an image, which amounts to around 1200 Kbits for the entire image of size 224x224. One must take into account that the concurrent transmission of even such small-resolution images from a large number of end devices impose a huge burden on the entire transmission network and power consumption. Our proposed method achieves a high **120x-240x** compression ratio, reducing the transmission cost to just 0.1-0.2 bits per pixel.
> > >
> > > * **[Inference Cost]** From Table 3 of our paper, when compared with a baseline method that decodes fully the image on the cloud side and uses a post processing network to enhance the compressed image for the MLLM, which requires 1017 kMAC/pixel, our latent adaptation technique has only 52 kMAC/pixel, i.e. a **reduction of nearly 95%** in kMAC/pixel.
> > >
> > > * **[Relation with Token Reduction for MLLM]** Notably, the reduction in the inference cost within MLLMs is NOT our focus. As mentioned, our proposed method is orthogonal to the techniques in this area (e.g. token reduction) and can be combined with them to streamline the entire system.
> > >
> > > Last but not least, as shown in Figure 3 and Figure 9 of our paper, our proposed method shows significant  rate-accuracy performance improvements as compared with the baseline methods that simply use the existing codecs (e.g. ELIC, VVC intra coding) to decode the image for the downstream MLLM tasks.
> > >
> > > ---
> > >
> > > **We have updated our paper**
> > > - **(L44-47 in Introduction)** to clarify the necessity of image compression.
> > > - **(L141-143 in Related work & Section A.6 of the supplementary document)** to address how our proposed method is related to token reduction techniques (and the like).
> > >
> > > Please see the text in blue color for changes.
> > >
> > > ---
> > >
> > > **We hope this response satisfactorily addresses your questions, and look forward to your favorable rating.**

---

> > > > ### Author Response · Authors · 2024-11-27
> > > > **Response to Reviewer HmZL**
> > > >
> > > > Thank you again for your thoughtful feedback on our work.
> > > >
> > > > We would like to note that we have further updated our paper according to your and other reviewers’ comments.
> > > > The following paragraph, highlighting the practical contributions of our work, is added to the end of Introduction (**L111-119**).
> > > >
> > > > > **Last but not least, the transform-neck trained with our surrogate loss exhibits a degree of universality, since it is readily applicable to multiple MLLMs that share the same visual encoder, without the need for retraining. Our method achieves (1) up to 60-80% bit-rate reductions under the same recognition accuracy over existing image codecs (e.g. ELIC (He et al., 2022) and VVC intra coding (Bross et al., 2021)) (Sections 4.2 and A.2) and (2) a nearly 95% reduction in decoding kMAC/pixel as compared to performing full image reconstruction followed by enhancing the reconstructed image for MLLM-based tasks (Section 4.4). Our system can be successfully trained under various application scenarios on one RTX 4090 with 24GB of memory. This is not possible when the entire MLLM is involved in the training process.**
> > > >
> > > > Additionally, we have updated the paper at:
> > > > - **L23-24 and L91-92** to highlight the versatility of our proposed method in all aspects.
> > > > > **The proposed framework is general in that it is applicable to
> > > > various MLLMs, neural image codecs, and multiple application scenarios. [...]**\
> > > > > **The proposed method is general in that it is applicable to different neural image codecs under various application scenarios.**
> > > > - **L82-87** to clarify the goal of our proposed method is not to develop a new and specific image codec but to adapt the compressed image latents of an existing neural image codec for MLLMs.
> > > > - **L184-185**  to note the task-specific nature of current mainstream coding for machines methods.
> > > > > **In addition, mainstream image coding for machines methods (e.g. Chen et al.
> > > > (2023b); Ascenso et al. (2023)) remain mostly task-specific.**
> > > > - **L482-483** to clarify the novelty of our proposed training objective.
> > > > > **Our proposed method, not a simple application of distillation, achieves the highest performance.**
> > > >
> > > > Please see the text in blue color for changes.
> > > >
> > > > We hope that our updated paper clarifies the contributions of the work. As the end of the discussion period is approaching fast, we would be happy to address any additional comments or make further revisions to the paper when needed.

---

> ### Author Response · Authors · 2024-11-25
> **Response to Reviewer HmZL**
>
> Dear Reviewer, thank you again for your thoughtful review of our paper. We have carefully addressed your concerns and comments in our previous rebuttal response to the extent possible. **As the end of discussion phase is fast approaching (Nov. 26 AoE)**, we look very much forward to your further comments (if any). If you find our responses satisfactory, we would kindly ask your consideration for increasing your rating on this pioneering attempt to perform image coding for MLLM-based tasks.

---

### Official Review · Reviewer_kXYn · 2024-11-01

**Soundness:** 3
**Presentation:** 3
**Contribution:** 3
**Rating:** 6
**Confidence:** 3

**Summary:**

The paper introduces a novel framework designed to bridge compressed image latents with Multimodal Large Language Models (MLLMs) for downstream vision tasks. It addresses the challenge of deploying MLLMs on resource-constrained devices by proposing a lightweight transform-neck and a surrogate loss to adapt compressed images for MLLM-based vision tasks without the need for decoding the images. This approach reduces computational complexity and is generic, applicable to various neural image codecs and MLLMs sharing the same visual encoder.

**Strengths:**

1. The paper introduces a lightweight transform-neck and a surrogate loss function, which together reduce computational complexity and avoid the need to back-propagate through the massive MLLMs, making the training process more efficient.
2. The proposed framework is generic and can accommodate various neural image codecs and MLLMs that share the same visual encoder, enhancing its versatility and applicability across different models and tasks.
3. The paper demonstrates the effectiveness of the proposed method through extensive experiments on different neural image codecs and MLLM-based vision tasks, showing improved rate-accuracy performance and less complexity compared to existing methods.

**Weaknesses:**

1. The paper's approach relies on the assumption that the downstream MLLMs will use the same pre-trained CLIP visual encoder, which may limit the applicability of the proposed method to MLLMs that employ custom or different visual encoders.
2. The paper does not provide a comprehensive comparison with existing image compression methods beyond the context of MLLMs, which could be important for understanding the method's performance in a broader range of applications

**Questions:**

1. Will you plan to release the code ?

---

> ### Author Response · Authors · 2024-11-20
> **Response to Reviewer kXYn**
>
> > **[W1] The paper's approach relies on the assumption that the downstream MLLMs will use the same pre-trained CLIP visual encoder.**
>
> **Our proposed framework is absolutely NOT restricted to specific visual encoders.** While our main experiments (Figure 3) are evaluated based on MLLMs that adopt the popular CLIP ViT visual encoder, the additional results in Section 4.6 and Figure 8 demonstrate the applicability of our method to two MLLMs with different visual encoders: (1) mPlug-Owl2 `[R1]` with a custom-trained ViT visual encoder, and (2) Osprey `[R2]` with a ConvNeXt-based visual encoder. We hope this clarification makes the matter clearer to the reviewer.
>
> ---
>
> > **[W2] The paper does not provide a comprehensive comparison with existing image compression methods beyond the context of MLLMs.**
>
> In the paper and supplementary document, we have provided the comparison with the existing image compression methods beyond the context of MLLMs, including ELIC `[R3]` (a learned image codec) and VVC Intra coding `[R4]` (the state-of-the-art traditional codec for intra image coding). Both are optimized for human perception. The former is labeled as "Reconstruction" in the performance comparison of Figure 3, and the comparison with the latter is included in Section A.2 of the supplementary document. **Our proposed method, tailored for MLLMs, outperforms both of these codecs significantly.**
>
> ---
>
> > **[Q1] Will you plan to release the code?**
>
> Yes, we shall release the source code upon the acceptance of the paper.
>
> ---
>
> **We thank the reviewer for the insightful comments.** We hope our responses have addressed the concerns satisfactorily, and would greatly appreciate that the reviewer considers increasing the rating on this pioneering attempt to perform image coding for MLLM-based tasks.
>
> ---
>
> ### **Reference**
> `[R1]` Q. Ye et al., "mplug-owl2: Revolutionizing multi-modal large language model with modality collaboration." arXiv, 2023. \
> `[R2]` Y. Yuan et al., "Osprey: Pixel understanding with visual instruction tuning." CVPR, 2024.\
> `[R3]` D. He et al., "Elic: Efficient learned image compression with unevenly grouped space-channel contextual adaptive coding." CVPR, 2022. \
> `[R4]` B. Bross et al., "Overview of the versatile video coding (vvc) standard and its applications." IEEE TCSVT, 2021.

---

> ### Author Response · Authors · 2024-11-25
> **Response to Reviewer kXYn**
>
> Dear Reviewer, thank you again for your thoughtful review of our paper. We have carefully addressed your concerns and comments in our previous rebuttal response to the extent possible. **As the end of discussion phase is fast approaching (Nov. 26 AoE)**, we look very much forward to your further comments (if any). If you find our responses satisfactory, we would kindly ask your consideration for increasing your rating on this pioneering attempt to perform image coding for MLLM-based tasks.

---

### Official Review · Reviewer_wKjV · 2024-11-05

**Soundness:** 2
**Presentation:** 3
**Contribution:** 2
**Rating:** 6
**Confidence:** 3

**Summary:**

This paper investigates adapting compressed image latents to meet the requirements of downstream vision tasks in Multimodal Large Language Models (MLLMs). The authors introduce a framework that employs a "transform-neck" structure and a surrogate loss function to adjust compressed image representations, without involving MLLMs in the training process. This design choice differs from conventional approaches that incorporate downstream models during training, making it potentially more feasible for resource-limited devices. The results show that this method achieves a balance between compression rate and accuracy across various neural image codecs and MLLMs, demonstrating its potential for efficient performance.

**Strengths:**

1. This paper focuses on an interesting scenario: how to achieve image compression in the context of MLLMs.
2. The writing in this paper is well-organized, making it easy to follow and understand the authors' intentions.

**Weaknesses:**

1.  The proposed method appears to be a general image compression approach, lacking unique design elements specifically tailored for MLLMs. While the authors emphasize that this approach can achieve image compression in a resource-efficient manner to support MLLMs, they do not provide specific comparisons regarding the reduction in training costs—such as training time or computational resources—compared to methods that incorporate MLLMs directly into the training process.
2. The proposed  SURROGATE LOSS is not novel, easily combination of the Mean Squared Error loss and the cross-entropy loss.
3. The proposed method has limited applicability, as it requires alignment with specific vision encoders. The authors only align with CLIP, whereas many advanced MLLMs now use other, more powerful vision encoders, such as Intern-ViT or SGLIP. This method may need further adaptation to support these alternative encoders effectively.

**Questions:**

1. Why are different MLLMs chosen for each task instead of evaluating multiple MLLMs on each task? A consistent comparison across tasks using various MLLMs would provide a clearer understanding of the method’s generalization capability.
2. During Phase 1: Transform-neck Training, the motivation for using different loss functions at various stages across all epochs is not clearly explained, and there is a lack of corresponding ablation studies to support this design choice.

---

> ### Author Response · Authors · 2024-11-20
> **Response to Reviewer wKjV (1/2)**
>
> > **[W1] The proposed method appears to be a general image compression approach, lacking unique design elements specifically tailored for MLLMs.**
>
> First, we like to clarify that it is NOT our intent to design a specific image compression system solely for MLLMs. As the title "*Bridging Compressed Image Latents and Multi-modal Large Language Models*'' of this paper suggests, we focus on how to **adapt the compressed image latents** produced by an existing image compression system to **suit the downstream MLLMs (and image reconstruction) tasks**. To this end, we have developed three application scenarios (see Figure 1 of the paper), two of which involve not only the MLLM task but also the image reconstruction task for human perception. Note that many real-world applications (e.g. surveillance) call for the flexibility that the compressed image latents be adapted to and used for both the machine and image reconstruction tasks. Our framework is able to enable this feature.
>
> Second, our system incorporates several novel components specifically for MLLMs: **(1) a surrogate loss** that combines the distillation loss for knowledge transfer and the cross-entropy loss to connect visual features to the text domain for MLLM-based tasks. Notably, for them to be effective, these two loss functions are applied via **(2) a progressive training strategy**. Last but not least, **(3)** the concept of **back-propagating the surrogate loss from the output of the visual encoder** that precludes large MLLMs in the training of our framework. To our best knowledge, these ideas are proposed for the first time to adapt the compressed image latents to MLLM-based tasks. We would be happy to discuss and cite any prior works from the reviewer that address similar issues.
>
> ---
>
> > **[W1] They do not provide specific comparisons regarding the reduction in training costs.**
>
> Our proposed framework enables efficient training with a surrogate loss back-propagated through only the visual encoder. In contrast, naively adopting the existing coding for machines techniques, which rely on back-propagating a task loss through the downstream recognition networks (MLLMs, in our case), would lead to extremely larger computational complexity. Taking a 7B MLLM model as an example, the sheer model size is **more than 20 times** that of the 300M CLIP visual encoder. With our limited compute resources, i.e. RTX 4090 with 24 GB of VRAM, we could not include such a large MLLM for end-to-end training.
>
> ---
>
> > **[W2] The proposed SURROGATE LOSS is not novel.**
>
> We respectfully disagree with the reviewer's statement regarding the lack of novelty of the surrogate loss. The training framework involving the proposed surrogate loss is illustrated in **[this figure](https://i.imgur.com/M5VIy2Q.png)**. To achieve high downstream MLLM performance, we propose an alternative to the straightforward yet undesirable approach of directly back-propagating a task loss through the entire MLLM. In fact, **our proposed surrogate loss is imposed at the output of visual encoder, with two distinct loss terms**: (1) the distillation loss for aligning the visual encoder output representations between using uncompressed image and transformed latents as input, and (2) the cross-entropy loss for bridging the visual features to the text domain. This dual-loss approach ensures effective latent transformation and has not been explored in prior work. Our finding is that relying solely on a single metric, such as the distillation or cross-entropy loss, leads to sub-optimal performance, as evidenced by our experiments in Section 4.5 of our paper.

---

> ### Author Response · Authors · 2024-11-20
> **Response to Reviewer wKjV (2/2)**
>
> > **[W3] The proposed method has limited applicability, as it requires alignment with specific vision encoders.**
>
> **Our proposed framework is absolutely NOT restricted to specific visual encoders.** While our main experiments (Figure 3) are evaluated based on MLLMs that adopt the popular CLIP ViT visual encoder, the additional results in Section 4.6 and Figure 8 of our paper demonstrate the applicability of our method to two MLLMs with different visual encoders: (1) mPlug-Owl2 `[R1]` with a custom-trained ViT visual encoder, and (2) Osprey `[R2]` with a ConvNeXt-based visual encoder. We hope this clarification makes the matter clearer to the reviewer.
>
> ---
>
> > **[Q1] Why are different MLLMs chosen for each task instead of evaluating multiple MLLMs on each task?**
>
> Our experiments involve multiple tasks with different MLLMs to show a wide spectrum of MLLM-task pairing for validating the effectiveness and generalizability of our proposed method. We believe our experiment setup is suitable to demonstrate this. As per the suggestion, we additionally provide the results of our proposed method on an additional task with two different MLLMs, Honeybee `[R3]` and Shikra `[R4]`, in Section A.5 of our revised supplementary document.
>
> ---
>
> > **[Q2] The motivation for using different loss functions at various stages across all epochs is not clearly explained.**
>
> Our training framework integrates the cross-entropy and distillation losses to achieve an effective latent transformation for downstream MLLMs. As demonstrated in Section 4.5 and Figure 6(a) of our paper, using either loss individually results in suboptimal performance. Additionally, Figure 7 of our paper provides further insights into the roles of each loss function: the cross-entropy loss focuses on minimizing the feature matching errors in the foreground object regions while the distillation loss is to minimize the global feature matching errors.
>
> ---
>
> **We thank the reviewer for the insightful comments.** We hope our responses have addressed the concerns satisfactorily, and would greatly appreciate that the reviewer considers increasing the rating on this pioneering attempt to perform image coding for MLLM-based tasks.
>
> ---
>
> **Reference**\
> `[R1]` Q. Ye et al., "mplug-owl2: Revolutionizing multi-modal large language model with modality collaboration." arXiv, 2023. \
> `[R2]` Y. Yuan et al., "Osprey: Pixel understanding with visual instruction tuning." CVPR, 2024.\
> `[R3]` J. Cha et al., "Honeybee: Locality-enhanced projector for multimodal llm." CVPR, 2024.\
> `[R4]` K. Chen et al., "Shikra: Unleashing multimodal llm’s referential dialogue magic." arXiv, 2023.

---

> ### Author Response · Authors · 2024-11-25
> **Response to Reviewer wKjV**
>
> Dear Reviewer, thank you again for your thoughtful review of our paper. We have carefully addressed your concerns and comments in our previous rebuttal response to the extent possible. **As the end of discussion phase is fast approaching (Nov. 26 AoE)**, we look very much forward to your further comments (if any). If you find our responses satisfactory, we would kindly ask your consideration for increasing your rating on this pioneering attempt to perform image coding for MLLM-based tasks.

---

### Official Review · Reviewer_ESiA · 2024-11-08

**Soundness:** 3
**Presentation:** 3
**Contribution:** 2
**Rating:** 6
**Confidence:** 4

**Summary:**

This paper suggests using compressed image latents for more efficient multimodal large language models (MLLMs). Specifically, a lightweight transform-neck and a surrogate loss are proposed to align compressed image latents with backbone LLMs for multimodal tasks. Experiments are conducted on some mainstream multimodal tasks, such as image captioning and VQA, and the proposed method demonstrates that it outperforms the reconstruction baseline.

**Strengths:**

* This paper is easy to follow
* The topic of reducing the cost of visual input in MLLMs is of great practical value
* The idea of adapting compressed image latent to MLLMs makes sense to me

**Weaknesses:**

This paper can be further strengthened by:

* One question that remains unclear about the motivation is why we need the proposed method of compressing image latent instead of existing token pruning/merging work on MLLMs such as crossget [1]. These methods can also reduce the costs of MLLMs, and I’d suggest the authors discuss the difference between existing acceleration methods for MLLMs and the proposed method, and highlight their unique contributions to efficient MLLMs.

* The proposed method shows inconsistent performance degradation on different benchmarks. For example, it shows close performance to models using uncompressed image latent on image captioning while suffering from a significant performance degradation on the POPE benchmark. It would be better to have some insight into it.

Reference

[1] Crossget: Cross-guided ensemble of tokens for accelerating vision-language transformers.

**Questions:**

Please refer to Weaknesses

---

> ### Author Response · Authors · 2024-11-20
> **Response to Reviewer ESiA**
>
> > **[W1] Why we need the proposed method of compressing image latent instead of existing token pruning/merging work on MLLMs such as crossget `[R1]`.**
>
> Token reduction methods (such as `[R1]`) differ fundamentally from our approach. They aim to reduce the number of visual tokens in MLLMs to lower computational complexity, and do NOT transmit visual tokens in compressed form. In contrast, our work addresses a fundamental preliminary step. Since the images must be transmitted to a cloud server for MLLM processing, we focus on **bridging the gap between compressed image latents and their suitability for downstream MLLM and/or image reconstruction tasks**.
>
> Ideally, one might propose generating visual tokens on the end device and using token reduction techniques as a compression method to reduce transmission bandwidth. However, this approach would impose significant computational demands on the end device, making it impractical for coding for machines scenarios, where the primary goal is to offload heavy feature extraction computations to the cloud.
>
> Notably, our method and the token reduction technique could potentially complement each other to develop a more efficient system. For instance, our approach allows to save transmission bandwidth and eliminates the need for image reconstruction, while token reduction techniques further optimize the computational complexity on the cloud side.
>
> ---
>
> > **[W2] The proposed method shows inconsistent performance degradation on different benchmarks.**
>
> The fact that the performance degradation varies across benchmarks/tasks is expected and has been observed in previous coding for machines literature [R2, R3]. These variations stem from the different sensitivity of the downstream task network to the image's compression artifacts. We stress that our proposed method outperforms the Reconstruction baseline across different tasks despite some variations, as shown in Figure 3 of our paper.
>
> ---
>
> **We thank the reviewer for the insightful comments.** We hope our responses have addressed the concerns satisfactorily, and would greatly appreciate that the reviewer considers increasing the rating on this pioneering attempt to perform image coding for MLLM-based tasks.
>
> ---
>
> **Reference**\
> `[R1]` D. Shi et al. "Crossget: Cross-guided ensemble of tokens for accelerating vision-language transformers," ICML 2024. \
> `[R2]` Y-H. Chen et al. "Transtic: Transferring transformer-based image compression from human perception to machine perception." ICCV, 2023. \
> `[R3]` H. Li et al., "Image compression for machine and human vision with spatial-frequency adaptation", ECCV 2024.

---

> ### Author Response · Authors · 2024-11-25
> **Response to Reviewer ESiA**
>
> Dear Reviewer, thank you again for your thoughtful review of our paper. We have carefully addressed your concerns and comments in our previous rebuttal response to the extent possible. **As the end of discussion phase is fast approaching (Nov. 26 AoE)**, we look very much forward to your further comments (if any). If you find our responses satisfactory, we would kindly ask your consideration for increasing your rating on this pioneering attempt to perform image coding for MLLM-based tasks.

---

> > ### Comment · Reviewer_ESiA · 2024-12-02
> >
> > Thank you for the rebuttal, which has addressed my concerns. I will keep my positive rating and suggest the authors carefully revise their paper to incorporate new discussions with reviewers.

---

### Author Response · Authors · 2024-11-20
**Response to All Reviewers**

We thank the reviewers for their insightful and constructive comments. Particularly, we encouraged by the reviewer (ESiA) for acknowledging the **practical value of our work**, and by reviewer (wKjV) for highlighting the **interesting scenario** of compression for MLLMs that our work addresses. We are also pleased that reviewers (kXYn, HmZL) appreciated the efficiency our proposed training objectives bring in terms of **computational complexity reduction**, and recognize the **versatility** of our framework, applicable across different MLLMs and practical scenarios.

We have carefully addressed all concerns and comments. We hope the reviewers would find our responses satisfactory and consider increasing their ratings on this pioneering attempt to perform image coding for MLLM-based tasks.

---

### Meta-Review · Area_Chair_AzUM · 2024-12-18

**Metareview:**

This paper studies using neural image compression to reduce the transmission rate between devices and servers for Multimodal Large Language Models. The proposed method involves a lightweight transform-neck, which is trained with surrogate loss to adapt image latents for MLLMs. The motivation of the paper is clearly explained and the paper is of good writing quality. The main weakness is the degradation on some benchmarks, which is also reported in previous literature. Given the detailed discussions of the methods in the paper and in the rebuttal, I would recommend accepting the paper.

**Additional Comments On Reviewer Discussion:**

Reviewers ESiA and wKjV asked about the differences between the proposed technique and the token pruning technique. The authors explain that these two methods are both effective but do not focus on the same aspect of compression.

Reviewer wKjV is also concerned about the limitation of the method to other vision encoders. The authors point out in the paper that they have experiments on other encoders in additional to CLIP ViT.

Reviewers HmZL and wKjV both asked for transmission cost and performance improvements. Reviewer kXYn asked for a comparison with other image compression techniques. The authors later added those in the rebuttal.

---

### Decision · Program_Chairs · 2025-01-22

Accept (Poster)